# Discriminant Canonical Tool for Differential Biometric Characterization of Multivariety Endangered Hen Breeds

**DOI:** 10.3390/ani11082211

**Published:** 2021-07-26

**Authors:** Antonio González Ariza, Ander Arando Arbulu, José Manuel León Jurado, Francisco Javier Navas González, Juan Vicente Delgado Bermejo, María Esperanza Camacho Vallejo

**Affiliations:** 1Department of Genetics, Faculty of Veterinary Sciences, University of Córdoba, 14071 Córdoba, Spain; angoarvet@outlook.es (A.G.A.); anderarando@hotmail.com (A.A.A.); juanviagr218@gmail.com (J.V.D.B.); 2Animal Breeding Consulting S.L., 14014 Córdoba, Spain; 3Agropecuary Provincial Centre, Diputación of Córdoba, 14071 Córdoba, Spain; jomalejur@yahoo.es; 4Andalusian Institute of Agricultural and Fisheries Research and Training (IFAPA), Alameda del Obispo, 14004 Córdoba, Spain; mariae.camacho@juntadeandalucia.es

**Keywords:** local breeds, genetic resources, biometric characteristics, phaneroptics, biodiversity

## Abstract

**Simple Summary:**

Breed undefinition boosts the risk of irreversible breed loss due to its substitution by dominant breeds. Breed loss results detrimental for the fraction of the genetic pool which is linked to the value of livestock as perfectly adapted elements of domestic ecosystems among other desirable features. In turn, this ensures and maximizes population sustainability. The present study aimed to design a biometric characterization tool in autochthonous avian breeds and their varieties in Andalusia (south of Spain): Utrerana and Sureña breeds. For this, different quantitative and qualitative measurements were collected in 473 females and 135 roosters belonging to these breeds. Even though both genotypes belong to a common original trunk, discriminant canonical analysis (DCA) revealed clear differences between both breeds and within the varieties that they comprise. In particular, certain variables such as ocular ratio and phaneroptic characteristics, which may be intrinsically related to the capacity of the breeds to adapt to the environmental conditions in which they thrive, could allow breeders to develop breeding programs focused on the enhancement productive potential of individuals.

**Abstract:**

This study aimed to develop a tool to perform the morphological characterization of Sureña and Utrerana breeds, two endangered autochthonous breeds ascribed to the Mediterranean trunk of Spanish autochthonous hens and their varieties (*n* = 608; 473 females and 135 males). Kruskal–Wallis H test reported sex dimorphism pieces of evidence (*p* < 0.05 at least). Multicollinearity analysis reported (variance inflation factor (VIF) >5 variables were discarded) white nails, ocular ratio, and back length (Wilks’ lambda values of 0.191, 0.357, and 0.429, respectively) to have the highest discriminant power in female morphological characterization. For males, ocular ratio and black/corneous and white beak colors (Wilks’ lambda values of 0.180, 0.210, and 0.349, respectively) displayed the greatest discriminant potential. The first two functions explained around 90% intergroup variability. A stepwise discriminant canonical analysis (DCA) was used to determine genotype clustering patterns. Interbreed and varieties proximity was evaluated through Mahalanobis distances. Despite the adaptability capacity to alternative production systems ascribed to both avian breeds, Sureña and Utrerana morphologically differ. Breed dimorphism may evidence differential adaptability mechanisms linked to their aptitude (dual purpose/egg production). The present tool may serve as a model for the first stages of breed protection to be applicable in other endangered avian breeds worldwide.

## 1. Introduction

In Spain, two hen trunks have historically been differentiated; the Atlantic trunk, generally comprising larger-format dual-purpose birds, with red earlobes and brown-shelled eggs, and the Mediterranean trunk, consisting of lighter individuals, with white earlobes and of a white-shelled egg-laying morphotype [1]. The aforementioned features have been considered by breeders on a regular basis for breed ascription and animal classification. This segregation of the Atlantic and Mediterranean trunks would later be supported from a molecular perspective through the estimation of genetic distances using microsatellite markers [2].

As a result, natural and human selection led to a high heterogeneity and variability of morphological characteristics in avian breeds [3,4]. Such high heterogeneity was promoted when breeding objectives (meat, eggs, or dual-purpose breeds) and, hence, morphological characteristics started to differ and polarize among populations to adapt to environment requirements at the minimum biological cost. These differentiation processes determined breeds to base their adaptability strategies on their particular enhanced body features [5].

Andalusia (Southern Spain) is influenced by the Mediterranean climate, with maximum temperatures rising above 40 °C in summer, as reported by the Spanish State Meteorological Agency (AEMET). In this context, very high temperatures are present from late spring on and last for the whole summer. Among the breeds in the area, two laying hen genotypes have traditionally configured poultry production under backyard and extensive systems: the Utrerana and Sureña avian breeds [6,7].

The Utrerana and Sureña avian breeds share a common geographic location, socioeconomic context, and history. In addition, four varieties of plumage color are present in both breeds: White, Franciscan, Black, and Partridge in the Utrerana breed; White, Franciscan, Black, Partridge, Blue, and Splash in the Sureña Breed. However, the Sureña hen has a larger format than most Mediterranean hen breeds [8,9].

These widely accessible low-capital/input investment birds were historically kept in sustainable systems for decades, thus becoming the source of production of high-biological-value proteins in rural livelihoods until globalization called for the intensification of animal production [10,11].

As a direct consequence, the population census of Spanish breeds suffered a regression due to the introduction of selected commercial strains of birds with a higher production during the last half of the 20th century [12,13]. In this way, the Utrerana avian breed became classified as an endangered breed, according to Royal Decree 45/2019 of 8 February, while the Sureña avian breed is in the process of being included in the Official Livestock Breeds Catalog of the Ministry of Agriculture, Fisheries, and Environment (MAPA) of Spain.

Consumers’ interest in quality food products revolved around market demands as a conscious response to the drawbacks implied by intensive production. Food alternatives produced through sustainable production systems became popular, provided these systems were characterized by a low impact on the environment and human health while they also considered animal welfare [14]. Increased demands soon translated into commercial chains starting to request differentiated products, whose properties significantly differed from products obtained through hybrid commercial strains [15].

For local producers to be able to fulfill market demands, products and the elements needed to ensure their constant supply must be defined through breed characterization zoometrically, genetically, or even productively. Contextually, the characterization of local populations, as well as the relationship among already established breeds, can provide pieces of evidence on the mechanism and events that contributed to the origin and development of native poultry breeds in the south region of Spain, as well as the adaptive mechanisms that may have permitted their survival in time [16]. Additionally, breed standardization could be an important tool for the evaluation of birds within their flocks and determine certain measurements for the selection of the best animals [17]. In this regard, morphometric and phaneroptic approaches may be fundamental in poultry management as they are fast and economically profitable [18].

This information altogether enables the correct development and implementation of the administrative structures needed to guarantee the stability and future viability of breeds through the development conservation and breeding programs, as well as the sustainable commercialization of their products once censuses are enough.

In this context, this study aimed to determine the contribution of quantitative and qualitative morphological-related traits to the zoometric characterization through the development of a discriminant canonical analysis (DCA), as a tool that permits determining phenotypic variability in the Andalusian avian breeds and within their varieties, as a strategy to support the standardization of native breeds and implement conservation strategies that ensure the consolidation of local genotypes as recognized breeds.

## 2. Materials and Methods

### 2.1. Animals, Sample Size, and Distribution

Biometric data were collected from 608 adult birds (from 1 to 7 years old, 1.94 ± 0.75 years), 473 hens (77.80%), and 135 roosters (22.20%), belonging to different varieties of Utrerana and Sureña breeds, as described in Figure 1. The sample size accounted for at least 20 times as many observations as variables. As this assumption was fulfilled, the study sample permitted to obtain reliable estimates of the canonical factor loadings for interpretation and to draw valid conclusions [19].

The sample was collected at 16 farms across the seven provinces in Andalusia (Cádiz, Córdoba, Granada, Huelva, Jaén, Málaga, and Sevilla). All animals were reared under extensive backyard conditions.

National guidelines for the care and the use of laboratory and farm animals, and avian-specific codes for good practices were followed during the data collection. For this, standards consistent with European Union legislation (2010/63/EU, from 22 September 2010) as transposed into Spanish law (Royal Decree Law 53/2013, from 1 February 2013). The study protocol was submitted to The Ethics Committee of Animal Experimentation of the University of Córdoba (Spain) and deemed exempt from review.

### 2.2. Biometric Measurement Collection

Biometrical analysis was performed in each bird, measuring 27 quantitative and five qualitative variables, following the procedure for morphological characterization of native chicken breeds described in previous studies [20,21]. A summary of the quantitative biometric variables and how to measure them is shown in Table 1. All corporal measurements were taken on the right side of the animal. Figure 2 shows details of the head measurements taken. A suspended electronic scale (measurement precision = 5 g; Kern CH50K100, Kern & Sohn, Balingen, Germany), a Vernier scale (Electro DH M 60.205, Barcelona, Spain), and a tape measure were used for measurement collection.

The following qualitative traits were evaluated in the present study: eye color, beak color, presence or absence of spurs, tarsus color, and nail color. Moreover, skull ratio, ocular ratio, beak ratio, and tarsus ratio were computed, as shown in Table 2.

### 2.3. Normality and Kruskall–Wallis Tests

The Shapiro–Francia W’ test (for 50 < *n* < 2500 samples) was used to discard gross violations of the normality assumption. The Shapiro–Francia W’ test was performed using the Shapiro–Francia normality routine of the test and distribution graphics package of the Stata Version 16.0 software (College Station, TX, USA). The normality test suggested normality assumption was not met. Hence, a nonparametric approach was followed. The Kruskal–Wallis H test was performed to detect differences in the median across sexes and genotypes. The Kruskal–Wallis H Test reported medians to significantly differ across all possibilities for sex and breed/variety combinations. Consequently, a separate DCA was performed for males and females.

### 2.4. Discriminant Canonical Analysis (DCA)

In the present research, 36 explanatory variables were used to perform the DCA: body weight, ornithological measurement, wingspan, skull length, skull width, ocular length, ocular width, beak length, beak width, comb length, comb width, number of spikes in the comb, earlobe length, earlobe width, wattle length, wattle width, neck length, back sternum length, tail length, thigh length, folding wing length, tarsus length, anteroposterior tarsus diameter, lateromedial tarsus diameter, eye color, beak color, presence or absence of spurs, tarsus color, nail color, skull ratio, ocular ratio, beak ratio, and tarsus ratio. In each sex, the breed and variety of the bird were used as classification criteria to measure the variability in morphological traits between and within the used classification groups and establish and outline population clusters [22,23].

The statistical analysis issued a set of discriminant functions that could be used as a tool to determine the clustering patterns described by the population sample through a linear combination of morphological-related traits. Furthermore, this canonical tool was used to plot pairs of canonical variables and graphically depict the group differences into an easily interpretable territorial map. Regularized forward stepwise multinomial logistic regression algorithms were used to perform the variable selection. Priors were regularized following the group sizes computed from the prior probability option in SPSS v26.0 software (IBM, Armonk, NY, USA), instead of considering them to be equal, thus preventing groups with different sample sizes from affecting the quality of the classification [24].

Previous studies have reported DCA to be robust and its outputs to be consistent when sample sizes among groups were highly unequal. Potential distortion effects derived from unequal sample sizing can be palliated using at least 20 samples for every four or five predictors. Additionally, the maximum number of independent variables must be *n −* 2 (where *n* = simple size). The present design was developed aiming at meeting these requirements sufficiently, to ensure the validity of the conclusions drawn.

Before discriminant analysis, independence of regressors was ensured by multicollinearity analysis. The same variables were chosen by the forward and the backward stepwise selection methods. Hence, the progressive selection method was chosen as preferable since it is less time-consuming than the backward selection method.

The discriminant routine of the Classify package of SPSS v26.0 software (IBM, Armonk, NY, USA) and the discriminant analysis routine of the analyzing data package of XLSTAT 2014 (Pearson Edition) (Addinsoft, Paris, France) were used to perform the DCA.

#### 2.4.1. Multicollinearity Preliminary Testing

Redundancies in the variables used were identified after performing the multicollinearity assumption before running the DCA. Multicollinearity analysis seeks to avoid the overinflation of the explanatory potential of variance due to the inclusion of an unnecessarily large number of variables. As an indicator of multicollinearity, the variance inflation factor was calculated using the following formula:(1)$$VIF=1/\left(1-R^{2}\right)$$
where *R*^2^ is the coefficient of determination of the regression equation.

A recommended maximum *VIF* value of 5 was used in the study, as suggested by Rogerson [25]. Tolerance (1 − *R*^2^) is the amount of variability in a certain independent variable that is not explained by the rest [26]. When tolerance values are lower than 0 and, simultaneously, *VIF* values ≥10, multicollinearity must be considered troublesome. *VIF* was computed using the discriminant analysis routine of the analyzing data package of XLSTAT 2014 (Pearson Edition).

#### 2.4.2. Canonical Correlation Dimension Determination

Pearson’s ρ was used to interpret canonical correlations. The maximum number of canonical correlations between two sets of variables is the number of variables in the smaller set. Although most of the relationships between different sets are explained by the first canonical correlation, all canonical correlations must be considered. Dimensions with canonical correlation values of ≥0.30 may be statistically significant.

#### 2.4.3. Discriminant Canonical Analysis Efficiency

Wilks’ lambda test was used to evaluate variables that significantly contribute to the discriminant function. When Wilks’ lambda approximates to 0, the contribution of the variable to a discriminant function increases. The chi-square statistic was considered to test the significance of Wilks’ lambda. If the significance is below 0.05, the function can be concluded to adequately explain the group adscription [27].

#### 2.4.4. Discriminant Canonical Analysis Model Reliability

Pillai’s trace criterion was used in the discriminant function analysis to test the assumption of equal covariance matrices. This is the only acceptable test that must be used in cases of unequal sample sizes [28]. Pillai’s trace criterion was calculated using the discriminant analysis routine of the analyzing data package of XLSTAT 2014 (Pearson Edition). A significance below 0.05 indicates significant statistical differences in the dependent variables across the levels of independence; hence, application of DCA is feasible.

#### 2.4.5. Variable Dimensionality Reduction

A preliminary principal component analysis (PCA) was computed to minimize overall variables into few meaningful variables that contributed to the morphological characterization of males and females in different genotypes. PCA was performed automatically using the discriminant analysis routine of the analyzing data package XLSTAT 2014 (Pearson Edition) (Addinsoft, Paris, France).

#### 2.4.6. Canonical Coefficient and Loading Interpretation and Spatial Representation

The percentage of allocation of an individual within its group (defined by its genotype) was calculated using a discriminant function analysis. Values ≥|0.40| in the discriminant loading of a variable were considered to be significantly discriminant. Thus, nonsignificant variables were excluded from the function using stepwise procedures. Higher values for absolute coefficients for each particular variable determine better discriminating power. Afterward, data were standardized following the premises reported by Manly and Alberto [29], and Mahalanobis distances were calculated using the following formula:(2)$$\;D_{ij}^{2}=({Ῡ}_{i}-{Ῡ}_{j})\;COV^{-1}({Ῡ}_{i}-{Ῡ}_{j})$$
where *D*^2^*_ij_* is the distance between population *i* and *j*, *Υ_i_* and *Υ_j_* are the means of variable *x* in the *i*-th and *j*-th populations, respectively, and *COV*^−1^ is the inverse of the covariance matrix of measured variable *x*. The squared Mahalanobis distance matrix was converted into a Euclidean distance matrix.

Afterward, dendrograms were built using the underweighted paired-group method arithmetic averages (UPGMA) from the Rovira i Virgili University, Tarragona, Spain, and the Phylogeny procedure of MEGA X 10.0.5 from the Institute of Molecular Evolutionary Genetics, The Pennsylvania State University, State College, PA, USA.

#### 2.4.7. Discriminant Function Cross-Validation

The percentage of correctly classified cases can be defined as the hit ratio. The leave-one-out cross-validation procedure was used to consider if the discriminant functions can be validated. Classification accuracy is achieved when the classification rate is at least 25% higher than obtained by chance.

*Press’s Q* statistic can support these results since it can be used to compare the discriminating power of the cross-validated function, as follows:(3)$$Press’s\;Q^{}=\frac{{\left[n-\left(n^{\prime }K\right)\right]}^{2}}{n\left(K-1\right)}$$
where *n* is the number of observations in the sample; *n*’ is the number of observations correctly classified, and *K* is the number of groups.

The value of the *Press’s Q* statistic must be compared with the critical value of 6.63 for χ^2^ with a degree of freedom in a significance of 0.01. When *Press’s Q* exceeds the critical value of χ^2^ = 6.63, the cross-validated classification can be regarded as significantly better than chance.

### 2.5. Data Mining CHAID Decision Tree

The chi-squared automatic interaction detection (CHAID) decision tree (DT) data mining method was used for classification, prediction, interpretation, and discrete categorized data manipulation. The tree routine of the Classify package of SPSS v26.0 software (IBM, Armonk, NY, USA) was used. Each internal node was built in the tree around a zoometric or phaneroptic trait (input variables), while a chi-squared test significance split criterion (*p* < 0.05 at least) was fulfilled in the so-called pre-pruning process.

Breiman, et al. [30] suggested that pre- or post-pruning methods prevent over-dimension of trees to prevent the failure to pursue the addition of traits (branches) which add significantly to the overall fit. As a result, a tree that exhaustively depicts the significant relationships across independent variables is one from which those nodes that do not contribute to the overall prediction have been discarded. Furthermore, CHAID additionally penalizes model complexity. In this regard, the Bonferroni inequality significant adjustment for significance levels was used.

Breiman’s method uses chi-squared tests to determine to configure the tree building process. Each branch represents an outcome of the test (in a number of two or more), and each leaf node (or terminal node) represents a category level of the target variable (breed/variety). The root node in the tree is the one that is located at the top. The decisions are made at each node, and each record of data continues through the tree along a path until the record reaches a leaf or terminal node of the tree [31].

Afterward, cross-validation was performed to validate the set of predictors considered measuring the differences between the prediction error for a tree applied to a new sample and a training sample. Cross-validation of the decision tree was performed using the “complexity parameter” and cross-validated error to estimate how accurately the model performs data prediction. Tenfold cross-validation [32] was performed using every sample record in the training sample and study data. The resubstitution error rate measures the proportion of original observations that were misclassified by various subsets of the original tree.

Tenfold cross-validation was used to obtain a cross-validated error rate, from which the optimal tree was selected to prevent bias and outlier overfitting. Tenfold cross-validation involves creating 10 random subsets of the original data, setting one portion aside as a test set, constructing a tree for the remaining (10 − 1) portions, and evaluating the tree using the test portion. This was repeated for all portions, and an estimate of the error was evaluated. Adding up the error across the 10 portions represented the cross-validated error rate. Afterward, the tree yielding the lowest cross-validated error rate was selected as the tree that best fit the data.

## 3. Results

### 3.1. Discriminant Canonical Analysis Reliability

Values of ρ < 0.05 obtained for Pillai’s trace criterion suggested the appropriateness of data to perform the DCA (Table 3). The contribution of canonical functions to the meaning of each discriminating function was assessed by Wilks’ lambda statistic (Table 4).

Appendix A show a summary of the values of tolerance and *VIF* for those variables for which *VIF* < 5 was reported and, thus, those which were included in the analysis across sexes. *VIF* values > 5 were discarded from further analyses; skull width, anteroposterior tarsus diameter, eye color, beak ratio, tarsus color, tarsus length, skull length, lateromedial tarsus diameter, and wingspan were the variables discarded for females, while lateromedial tarsus diameter, ocular width, skull width, beak ratio, nail color, tail length, eye color, tarsus color, wattle width, tarsus length, and skull length were the traits discarded before DCA in male individuals.

### 3.2. Canonical Coefficients, Loading Interpretation, and Spatial Representation

DCA determined three discriminating canonical functions for both sexes (Table 4 and Table 5). Lower Wilks’ lambda values and respective higher eigenvalues were indicative of higher discriminating power. In females, 90.37% of the total variance was explained by functions F1 and F2 (eigenvalues of 9.66 and 5.17 for F1 and F2, respectively). In males, functions F1 and F2 (eigenvalues of 26.91 and 7.34 for F1 and F2, respectively) explained 88.49% of the total variance.

After discarding redundant variables, variables were ranked by the test of equality of group means across groups depending on their discriminating properties (Table 6 and Table 7). Lower values of Wilks’ lambda and greater values of F indicated a better discriminating power, which translated into a better position in the rank.

Figure 3 presents a graph of standardized discriminant coefficients across discriminant functions. These analyses not only allowed us to easily identify those variables accounting for higher repercussions on the discriminant power of functions overall, but also the possibility of a reduction in the discriminant power of individual variables as a result of multicollinearity between pairs.

The substitution of the values for biometric-related traits into the first three discriminating functions was performed to obtain *x-*, *y-*, and *z*-axis coordinates, for the first, second, and third dimensions, respectively. In these coordinates, each observation was sorted and classified across the different groups. A territorial map was depicted for each sex (Figure 4).

Mahalanobis distance represents the probability that an observation presenting an unknown background belongs to a particular group (breed/variety). It can be computed through the relative distance of the problem observation to the centroid of its closest group. Then, the hit ratio was calculated. The hit ratio is the rate of successfully classified cases across breed/varieties (which was performed across sexes) (Appendix A). Mahalanobis distances obtained after the evaluation of the discriminant analysis matrix were transformed into squared Euclidean distances, and the results are represented in Figure 5 and Figure 6, following Hair et al. [33].

Appendix A report the results obtained in the classification and leave-one-out cross-validation for the observations in the present study. Here, 71.82% and 81.48% of original grouped cases were correctly classified for females and males, respectively. From these results, 59.96% and 49.63% of clustered observations were cross-validated. Press’s Q values of 2004.41 and 1060.27 were obtained from females and males, respectively; hence, it can be considered that predictions were significantly better than chance at 95% [34].

### 3.3. Data Mining CHAID Decision Tree

The underlying basis for these classification patterns was found after the evaluation of the data mining CHAID decision tree obtained for the chi-square dissimilarity matrix. Classification trees of groups by genotypes produced simple trees with terminal nodes (Appendix A). Chi-squared-based branch and node distribution suggested females significantly (*p* < 0.001) differed depending on their values of nail color and, thus, were classified into four subgroups (black corneous/slate corneous, slate, corneous, and white). Nail color was the best discriminant phaneroptic trait and helped to distinguish among black Utrerana, black Sureña, Partridge Utrerana, and Franciscan Utrerana). Afterward, ocular ratio helped to discriminate across the varieties of Utrerana and Sureña hens (*p* < 0.001), with the Utrerana animals presenting ocular indices over 0.986, while Sureña ocular indices were equal to or below 0.986 (Appendix A).

By contrast, chi-squared-based branch and node distribution suggested males only significantly (*p* < 0.001) differed depending on their values of ocular ratio. Ocular ratio helped to discriminate between varieties of Utrerana and Sureña roosters (*p* < 0.001), with the Utrerana animals presenting ocular indices over 1.015, while Sureña ocular ratios were equal to or below 1.015 (Appendix A).

Female data mining decision tree tenfold cross-validation reported closely similar resubstitution (probability of misclassifying an unseen instance) and cross-validation error rate estimates of 0.484 and 0.510, for which the standard error was 0.023, respectively. For the male tree, 0.726 and 0.867 values of resubstitution and cross-validation error rate estimates were obtained with standard errors of 0.038 and 0.029, respectively. Although data resubstitution can underestimate the classifier error, it has less variability than other methods, such as cross-validation, especially for small sample sizes. As cross-validation error rate estimates were close to resubstitution ones, albeit lower, trees were not overfitted, confirming the robustness of the results obtained and the validity of the conclusions drawn.

## 4. Discussion

Differential sex-linked hormonal and genetic regulation patterns of the expression of growth have been reported to occur in local poultry breeds [35,36]. Dimorphism and dichromatism could be a consequence of sexual selection and might provide an adaptative advantage of one population over others. For instance, in the context of the conditions found in rustic backyard environments, even if there is a lower selective pressure focused toward production, male-to-male competition has induced roosters to increase the size, giving an advantage against the opponent [37].

In the context of multizoometric and phaneroptic analyses, it has been suggested that it is necessary to check for the different relationships across explanatory variables and select independent variables that do not overlap when deciding on the factors which determine the efficiency of predictive models [23]. High correlations between skull length and skull width (i.e., skull ratio) were revealed by the multicollinearity analysis since the formula for skull ratio calculation comprises the aforementioned measurements. The same happened with anteroposterior (in both sexes) and lateromedial (only in hens) tarsus diameters as the elements which determine the tarsus ratio. The calculation formula of beak ratio, which includes the remaining beak measurements, was eliminated from further analysis due to multicollinearity problems (*VIF* > 5).

Lastly, the ocular width variable was discarded from the analysis of male individuals since this variable is contained within the formula of ocular ratio (*VIF* > 5). These results are supported by those in Ning et al. [38], who found multicollinearity problems when formulae were developed after the inclusion of explanatory variables which were already included.

Phaneroptic variables have been reported to be highly significantly interrelated [39]. Even if most qualitative variables were discarded after the multicollinearity analysis, nail color in hen and beak color in roosters were the only qualitative variables that remained in the DCA. Thus, results suggest that multicollinearity problems between different qualitative measurements in birds may have occurred.

White nails was reported to be the best discriminating feature in hens (Table 6). Only seven individuals of White, Splash, and Franciscan Sureña showed dark nails, while no hen of White and Franciscan varieties showed nails of a different color than white. In roosters, black/corneous and white colors in the beak were also reported to have high discriminant power.

Previous studies have reported that phaneroptic features are somehow correlated in native chicken breeds, provided they may derive from the expression of the same gene background across the body parts [40]. Additionally, it has been suggested that these qualitative traits have significant effects on other quantitative traits such as body weight and daily gain in chicken [40,41].

Our results are indicative of the fact that qualitative variables, with high discriminant ability to discern among local hen genotypes, must be considered as efficient selection criteria in breeding programs, as an effective method to identify the individuals presenting the most desirable production-related characteristics at the most convenient earlier age.

Furthermore, certain phaneroptic variables may be associated with consumers’ trends and their cultural preferences. For instance, while North American consumers have strong preferences for white-skin meat [42], meat from dark-skin poultry is preferred by producers and consumers in South America [14]. Hence, multivariety breeds accounting for a wide variety of feather and skin color patterns such as Utrerana and Sureña could satisfy the needs of a wider scope of targets in different market niches.

Feather coloration strongly conditions the camouflage abilities of birds. In this regard, Dohner [43] suggested that the less aggressive strains developed for confinement may be less self-sufficient and may not be as alert to predators. In hens, this has been ascribed to the association of specific quantitative trait loci with behavioral traits [44]. As an example, birds carrying the ancestral junglefowl allele (i) of the *PMEL17* locus are black, while White Leghorn (I) birds are white (with heterozygotes frequently being less pigmented).

Contextually, *i*/*i* alleles carriers have been reported to be more vocal, less prone to develop fearful attitudes toward humans, and more aggressive, social, and explorative (enhanced foraging behavior) [44]. These enhanced behavioral features may make these dark-colored breeds less susceptible to predation by hawks [43]. The *PMEL17* locus has simultaneously been associated with feather-pecking and bullying behavior toward counterparts [45], with darker birds tending to be rather affected by feather-pecking than their white counterparts [46]. It is still unknown whether feather-pecking may exclusively be attributed to plumage color or to the behavior of *i*/*i* carrier individuals to become targets of pecking attacks.

Alternatively, Tickell [47] stated that coloration-related costs in higher rates of bird predation may also translate into the enhancement of other tactics for evading capture [6]. This was reflected in our study (Figure 5 and Figure 6) with Sureña presenting smaller ocular indices in comparison to Utrerana hens, albeit with darker Sureña individuals being closer to white Utrerana animals and white-feathered Sureña located further away when morphological traits were considered.

Ocular ratio was ranked second and first regarding its discriminant ability in hens and roosters, respectively. The relevance of ocular ratio may be ascribed to higher adaptability to the environment and improved capacity to seek food as a result of improved vision skills. Indeed, except for certain occasions, birds have a highly developed vision.

In relationship to the size of the skull, the avian eye is very large. While humans have an eye relative size of 5% with respect the skull, in hens, 50% of the cranial volume is occupied by the orbit [48]. High visual acuity is advantageous for hens relying heavily on their ability to navigate surroundings to find and acquire food, to identify potential mates, and to quickly escape from predators [49,50]. Hall and Ross [51] reported that the light level, which is highly correlated with bird activity pattern, has a more significant influence on eye shape and body size than other factors, such as phylogeny.

Birds with a higher adaptation to darkness habits, such as brooding and nesting abilities, exhibit larger axial and corneal lengths and, therefore, a higher eye size diameter than the rest of the birds [52,53]. On the other hand, larger individuals with larger eyes have the potential for more sensitive and acute vision than smaller individuals with smaller eyes. This could suggest that the Sureña breed, with a significantly larger eye size, has a sharper vision. However, each breed has developed an ideal eye design for conditions in which it is produced. Larger eyes need more brain space for information processing. Therefore, evaluation of ocular size in each breed must be performed taking into account body size [54]. Thus, the higher size of Sureña eyes could be mainly ascribed to a proportionally larger body shape.

It has also been suggested that lower values for ocular ratio may act as an adaptation to optimal antipredator behavior since larger ocular width could suppose an advantage in the lateral visual field [55,56]. Thus, results obtained in the present study may suggest that Utrerana eyes make it more adapted to survival in free-range systems. Furthermore, smaller birds have developed rather improved adaptative qualities such as hardiness, agility, scavenging ability, and less time needed for flight [57]. The Utrerana breed, with lower body weight and ocular ratio, may be better adapted to free-range systems through its enhanced rusticity, even if the literature indicates that both breeds can easily thrive and are well adapted to the environmental conditions present in these alternative production systems [9,58].

Back length was the third best discriminant variable in hens. These results agree with those presented by previous research [59,60]. In this sense, back length has been reported to be highly correlated with other important traits. As a consequence, it plays an important role as a linear body measurement when the aim is to predict for body weight, as well as to develop and to implement productive selection strategies during breeding in laying hens.

Size-related parameters such as body weight (in hens) and wingspan (in roosters) play a pivotal role in the classification of individuals (Table 6 and Table 7). These traits allow us to delimitate those animals belonging to the Sureña breed. Sureña individuals typically account for larger body sizes than Utrerana individuals.

Lighter hens have been reported to present higher egg productions and lower feed conversion rates and, therefore, a better laying ability [61]. On the other hand, breeds characterized by larger individuals may be prone to become dual-purpose genotypes in alternative production systems, in which both sexes are reared together, to later, at an advanced age, separate males for final fattening and slaughtering, while females are kept during several laying cycles [62,63]. Bearing this in mind, focusing efforts on the selection of the Utrerana breed toward an egg production aptitude and Sureña as a dual-purpose breed may be the most effective and profitable productive alternative.

Although Sureña and Utrerana breeds were presumably selected from a common origin [8], the graphic representation of the observations assessed in the present study (Figure 4) reports a clear differentiation of morphological characteristics between the two breeds. While three clear clusters are shown in Utrerana breed (Partridge, Black, and Franciscan/White varieties), the closeness of the six varieties of the Sureña avian breed suggests a likely lack of reproductive management and crossbreeding among the different varieties of this breed.

This proves that, once official breed recognition occurs, an incorrect application of a breeding program in local breeds can lead to a deterioration of the phenotypic and genotypic identity of the individuals, which directly results in the partial or total loss of the genetic pool of these local resources [64,65].

Contextually, Partridge Utrerana was reported to be the most differentiated variety from all studied varieties. These results are supported by those in Macrì et al. [6], who reported Partridge Utrerana individuals to be placed the farthest away from the rest of Utrerana varieties.

More than 75% of hens in each Utrerana variety were correctly classified (Appendix A), except for the individuals of the White variety, whereby 50% of hens were notably classified as Franciscan Utrerana hens. This Utrerana White/Franciscan misclassification is supported by the results in Figure 5 and Figure 6. Franciscan and White Utrerana varieties were closely clustered (Figure 5 and Figure 6). This finding may indirectly indicate reminiscences of hybridization between White and Franciscan Utrerana varieties, with both presenting white legs and beak, which may be the result of the attempts of breeders to decrease the consanguinity within the White Utrerana variety, given that this variety has historically been the subpopulation accounting for the smallest census and that which faces the highest endangerment risk [22].

Blue Sureña variety females were those for which a rather frequent misclassification rate occurred (Appendix A). This finding may stem from the fact that breeding practices performed in the area may seek the obtention of individuals presenting blue plumage patterns through a cross between other varieties, such as Black or Splash [66].

Biometric studies have been performed worldwide to make breed characterization feasible and to be considered during the implementation of conservation strategies and policies [16]. This suggests that the preservation of the breed diversity may be one of the motor elements ensuring the future survival of a breed. This future survival may rely on the enhancement of a breed’s ability to cover a wider scope of market demands, thereby reaching a broader audience [67]. The present methodological proposal is framed in the context of opportunity and resurgence of a potential production industry that intends to lay the base for a sustainable selective breeding program in avian breeds. Certain easily measurable traits, such as phaneroptic variables and ocular ratio, can efficiently play a pivotal role in the classification of birds. In this context, the discriminant tool designed in the present research allows efficiently classifying individuals considering biometric and phaneroptic traits. This is supported by the 71.82% and 81.48% of individuals correctly ascribed to their prior hen breed/variety cluster.

## 5. Conclusions

Sexual selection of larger males in backyard production systems may evidence clear sexual dimorphism in Utrerana and Sureña breeds. The use of these multivariate breeds is productively advantageous since a broader scope of market demands could be satisfied in terms of carcass organoleptic characteristics. This research confirms that native breeds in the south of Spain may be well adapted to extensive and backyard systems, but also that their differential zoometric adaptation may make them more suited for the aptitude that they were selected to perform. Nevertheless, the Utrerana breed showed a better morphological adaptation to optimal antipredator behavior and rusticity. In any case, both breeds should follow different breeding programs considering alternative routes; the Sureña breed has greater potential as a dual-purpose breed, while morphometric traits of the Utrerana breed may be indicative of higher profitability in egg-producing farms. The present research validates the efficiency of the discriminant tool designed while performing individual selection and breed ascription considering easily measurables traits such as ocular ratio and phaneroptic variables, which may simultaneously ensure the survival of these local genetic resources.

## Figures and Tables

**Figure 1 animals-11-02211-f001:**
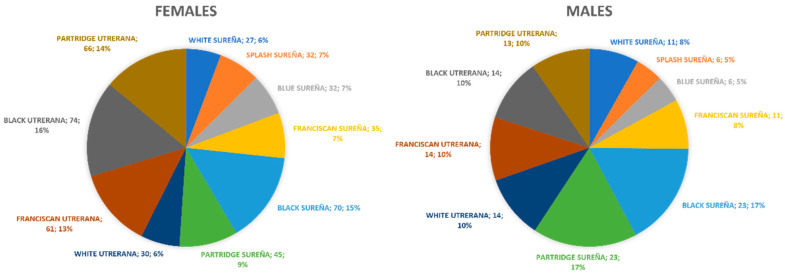
Percentage and number of individuals (*n*) used in each studied genotype.

**Figure 2 animals-11-02211-f002:**
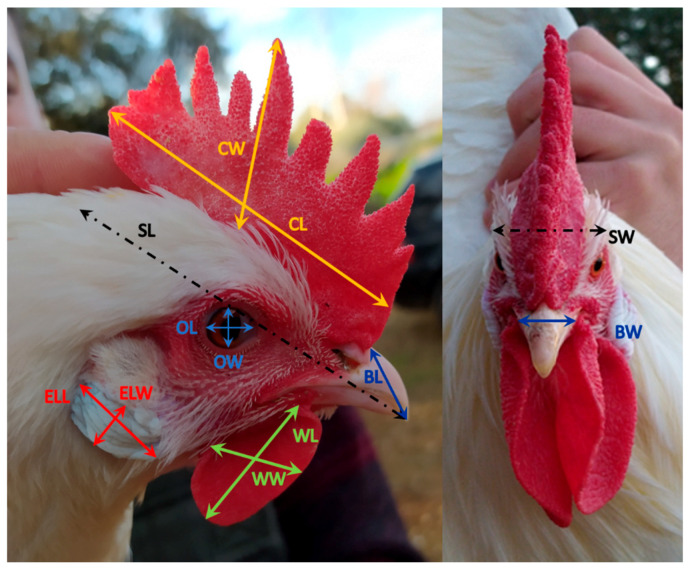
Detailed views of a hen and a rooster head with their corresponding measures. CL: comb length, CW: comb width, OL: ocular length: OW: ocular width, SL: skull length, SW: skull width, BL: beak length, BW: beak width, ELL: earlobe length, ELW: earlobe width, WL: wattle length, WW: wattle width.

**Figure 3 animals-11-02211-f003:**
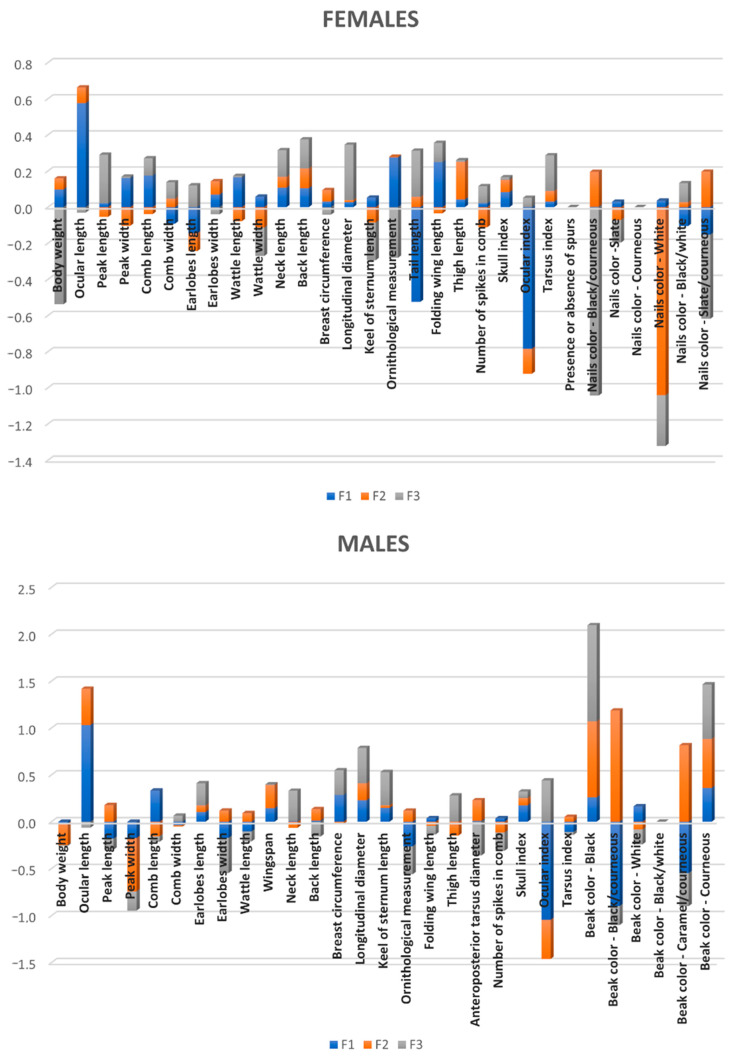
Discriminant loadings for biometric quality-related traits determining the relative weight of each trait on each canonical discriminant function. Each bar represents the relative weights (coefficients) of each variable across the three discriminant functions revealed by CDA.

**Figure 4 animals-11-02211-f004:**
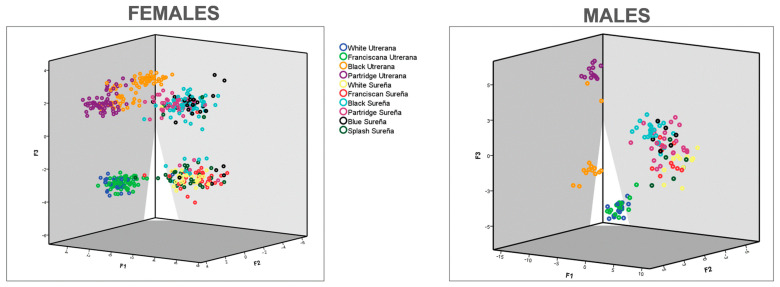
Territorial map depicting the observations considered in the canonical discriminant analysis sorted across genotypes.

**Figure 5 animals-11-02211-f005:**
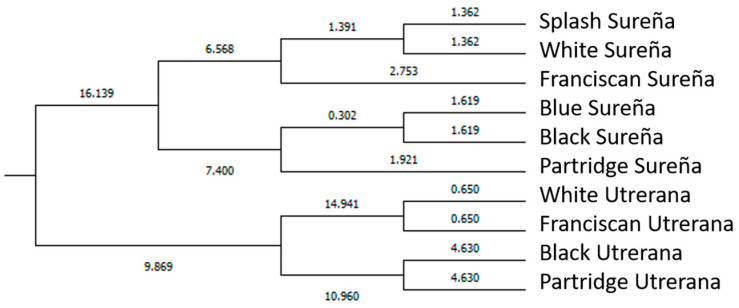
Cladogram constructed from Mahalanobis distances across different genotypes (breed/varieties) in female population.

**Figure 6 animals-11-02211-f006:**
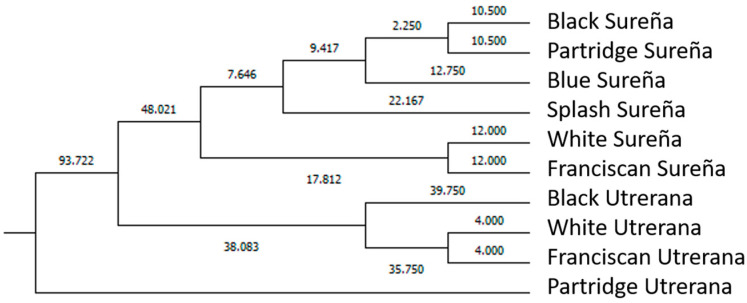
Cladogram constructed from Mahalanobis distances across different genotypes (breed/varieties) in male population.

**Table 1 animals-11-02211-t001:** Biometric variables and measuring procedures used in the present study.

Corporal Region	Variable	Units	Measuring Procedure
General characteristics	Bodyweight	kg	With an electronic scale
	Ornithological measurement	cm	Leaning the bird on its back, the distance between the tip of the beak and the tip of a central rectrix, in a straight line
	Wingspan	cm	Distance between the ends of the longest primaries with outstretched wings
Head	Skull length	mm	Taken between the most protruding point of the occipital and the tip of the beak
	Skull width	mm	Taken at eye level
	Comb length	mm	Measured between the insertion of the comb in the beak and the end of the comb’s lobe
	Comb width	mm	Measured from the tip of the central spike until the insertion of the comb in the skull; when the number of spikes was even, the highest was chosen
	Number of spikes in the comb	*n*	By manual counting
	Ocular length	mm	Measured between eyelid corners
	Ocular width	mm	Measured including the folds of the eyelid, perpendicular to the ocular length
	Beak length	mm	Measured from the tip of the beak until the insertion of the beak in the head
	Beak width	mm	Measured at level of insertion of the beak in the head
	Earlobe length	mm	Maximum length, keeping the bird’s head perpendicular to the neck
	Earlobe width	mm	As in the previous measure, measured the second-largest dimension
	Wattle length	mm	Measured from the insertion of wattle in the beak until the end of the wattle, in a straight line
	Wattle width	mm	As in the previous measure, measured the second-largest dimension
Neck	Neck length	cm	Distance from the base of the neck to the chest
Body	Back length	cm	Distance from the insertion of the neck into the body to the tail insertion
	Keel of sternum length	cm	Leaning the bird on its back, the distance between the two vertices of the sternum
	Breast circumference	cm	Measured at the level of the tip of the keel, passing the tape measure through the back of the wing insert
	Longitudinal diameter	cm	Measured from the cranial end of the coracoid to the most caudal portion of the pubis
	Tail length	cm	Distance from the tip of a central rectrix to the insertion of the tail
Extremities	Folding wing length	cm	Distance from the carpal joint until the end of the longest primary
	Thigh length	cm	Distance from the middle region of the coxal bone to the knee joint
	Tarsus length	cm	Distance from the notch of the shinbone tarsus until the tip of the nail of the middle finger
	Anteroposterior tarsus diameter	mm	Diameter of the tarsus in an anteroposterior direction in the middle part of the metatarsus bone
	Lateromedial tarsus diameter	mm	Diameter of the tarsus in a lateromedial direction in the middle part of the metatarsus bone

**Table 2 animals-11-02211-t002:** Mathematical description of biometric indices.

Trait	Mathematical Expression	
Skull ratio	SI=SL/SW	*SI*: skull ratio; *SL*: skull length; *SW*: skull width
Ocular ratio	OI=OL/OW	*OI*: ocular ratio; *OL*: ocular length; *OW*: ocular width
Beak ratio	BI=BL/BW	*BI*: beak ratio; *BL*: beak length; *BW*: beak width
Tarsus ratio	TI=APTD/LMTD	*TI*: tarsus ratio; *APTD*: anteroposterior tarsus diameter; *LMTD*: lateromedial tarsus diameter

**Table 3 animals-11-02211-t003:** Summary of the results of Pillai’s trace of equality of covariance matrices of canonical discriminant functions.

Females	Pillai’s trace criterion	2.8664
	F (Observed value)	7.1227
	F (Critical value)	1.1540
	df1	261
	df2	3978
	*p*-value	<0.0001
	alpha	0.05
Males	Pillai’s trace criterion	3.8256
	F (Observed value)	2.7989
	F (Critical value)	1.1740
	df1	252
	df2	954
	*p*-value	<0.0001
	alpha	0.05

F, Snedecor’s F; df1, numerator degrees of freedom for the F-approximation (groups minus 1); df2, denominator degrees of freedom for the F-approximation (observations minus 1).

**Table 4 animals-11-02211-t004:** Canonical discriminant analysis efficiency parameters to determine the significance of each canonical discriminant function.

	Test of Function(s)	Wilks’ Lambda	Chi-Square	df	Sig.
Females	1 through 7	0.045	1436.63	63	0
	2 through 7	0.411	410.85	48	0
	3 through 7	0.814	95.218	35	0
Males	1 through 4	0.017	515.527	36	0
	2 through 4	0.242	180.18	24	0
	3 through 4	0.813	26.252	14	0.024

**Table 5 animals-11-02211-t005:** Canonical variable functions and percentage of self-explained and cumulative variance.

Sex	Function	Eigenvalue	Discrimination (%)	Cumulative %
Females	F1	9.6611	58.8681	58.8681
	F2	5.1701	31.5034	90.3716
	F3	0.7705	4.6950	95.0665
Males	F1	26.9110	69.5353	69.5353
	F2	7.3362	18.9561	88.4914
	F3	2.7997	7.2342	95.7256

**Table 6 animals-11-02211-t006:** Results for the tests of equality of females group means to test for difference in the means across groups once redundant variables were removed in the female population.

Variables	Lambda	F	df1	df2	ρ-Value	Rank
Nail color (white)	0.1911	217.2864	9	462	<0.0001	1
Ocular ratio	0.3571	92.3999	9	462	<0.0001	2
Back length	0.4291	68.3067	9	462	<0.0001	3
Body weight	0.4318	67.5522	9	462	<0.0001	4
Ocular length	0.4982	51.6983	9	462	<0.0001	5
Longitudinal diameter	0.5184	47.6874	9	462	<0.0001	6
Keel of esternum length	0.5262	46.2222	9	462	<0.0001	7
Wattle length	0.5381	44.0615	9	462	<0.0001	8
Folding wing length	0.5691	38.8630	9	462	<0.0001	9
Comb length	0.5828	36.7513	9	462	<0.0001	10
Wattle width	0.5986	34.4272	9	462	<0.0001	11
Breast circumference	0.6052	33.4926	9	462	<0.0001	12
Thigh length	0.6358	29.4067	9	462	<0.0001	13
Nail color (black/corneous)	0.6736	24.8741	9	462	<0.0001	14
Ornithological measurement	0.6831	23.8125	9	462	<0.0001	15
Comb width	0.6868	23.4102	9	462	<0.0001	16
Beak width	0.6935	22.6921	9	462	<0.0001	17
Earlobe width	0.7001	21.9939	9	462	<0.0001	18
Tail length	0.7660	15.6822	9	462	<0.0001	19
Beak length	0.7855	14.0167	9	462	<0.0001	20
Earlobe length	0.8005	12.7947	9	462	<0.0001	21
Nail color (slate/corneous)	0.8156	11.6036	9	462	<0.0001	22
Nail color (slate)	0.8426	9.5928	9	462	<0.0001	23
Skull length	0.8629	8.1568	9	462	<0.0001	24
Number of beaks in comb	0.9095	5.1094	9	462	<0.0001	25
Tarsus ratio	0.9416	3.1857	9	462	0.0009	26
Skull ratio	0.9703	1.5692	9	462	0.1217	27
Nail color (black/white)	0.9869	0.6793	9	462	0.7279	28
Presence or absence of spurs	0.9903	0.5005	9	462	0.8743	29

F, Snedecor’s F; df1, numerator degrees of freedom for the F-approximation (groups minus 1); df2, denominator degrees of freedom for the F-approximation (observations minus 1).

**Table 7 animals-11-02211-t007:** Results for the tests of equality of group means test for difference in the means across groups once redundant variables were removed in the male population.

Variables	Lambda	F	df1	df2	ρ-Value	Rank
Ocular ratio	0.1797	63.4040	9	125	<0.0001	1
Beak color (black/corneous)	0.2102	52.1922	9	125	<0.0001	2
Beak color (white)	0.3489	25.9192	9	125	<0.0001	3
Wingspan	0.3765	22.9996	9	125	<0.0001	4
Beak color (black)	0.4526	16.7993	9	125	<0.0001	5
Back length	0.4547	16.6534	9	125	<0.0001	6
Ocular length	0.5279	12.4222	9	125	<0.0001	7
Longitudinal diameter	0.5536	11.1984	9	125	<0.0001	8
Anteroposterior tarsus diameter	0.5576	11.0173	9	125	<0.0001	9
Body weight	0.6399	7.8142	9	125	<0.0001	10
Breast circumference	0.6511	7.4427	9	125	<0.0001	11
Folding wing length	0.6653	6.9859	9	125	<0.0001	12
Earlobe width	0.7245	5.2821	9	125	<0.0001	13
Beak color (corneous)	0.7272	5.2092	9	125	<0.0001	14
Keel of sternum length	0.7424	4.8184	9	125	<0.0001	15
Wattle length	0.7819	3.8731	9	125	0.0002	16
Comb length	0.7899	3.6936	9	125	0.0004	17
Beak width	0.7903	3.6848	9	125	0.0004	18
Beak length	0.8000	3.4712	9	125	0.0007	19
Earlobe length	0.8194	3.0609	9	125	0.0024	20
Number of beaks in comb	0.8225	2.9981	9	125	0.0029	21
Thigh length	0.8296	2.8519	9	125	0.0043	22
Neck length	0.8707	2.0623	9	125	0.0378	23
Ornithological measurement	0.8798	1.8980	9	125	0.0580	24
Comb width	0.9029	1.4932	9	125	0.1574	25
Tarsus ratio	0.9072	1.4215	9	125	0.1858	26
Skull ratio	0.9254	1.1189	9	125	0.3544	27
Beak color (caramel/corneous)	0.9300	1.0460	9	125	0.4077	28

F, Snedecor’s F; df1, numerator degrees of freedom for the F-approximation (groups minus 1); df2, denominator degrees of freedom for the F-approximation (observations minus 1).

## Data Availability

All data stemming from the present research has been enclosed in tables or as Appendix A. Any additional data will be made accessible from corresponding authors upon reasonable request.

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
