# Peer review of "Discriminant Canonical Tool for Differential Biometric Characterization of Multivariety Endangered Hen Breeds"

_animals, 2021, doi:10.3390/ani11082211_

Round 1

Reviewer 1 Report

This is a very well written and interesting manuscript designing a biometric characterization tool in autochthonous avian breeds and their varieties in Andalusia. This has certain significance for the protection of local poultry and the improvement of production performance. There are few minor discrepancies that need to be addressed:

  1. Simple Summary is relatively tedious.Please simplify it and do not repeat it with the introduction.
  2. In the part of materials and methods, 608 1-7 adult birds were selected as experimental materials.Whether it has some reference value for other commercial breeds or models aiming at obtaining the maximum economic value in a short time.
  3. The clarity of figure 1, 3, and 4 is low, please change them.
  4. Some of the references are formatted incorrectly. Please review them and modify them according to the format of Animals.

Author Response

REVIEWER 1

This is a very well written and interesting manuscript designing a biometric characterization tool in autochthonous avian breeds and their varieties in Andalusia. This has certain significance for the protection of local poultry and the improvement of production performance.

Response: We thank the reviewer for his/her kind comments.

There are few minor discrepancies that need to be addressed:

  1. Simple Summary is relatively tedious. Please simplify it and do not repeat it with the introduction.

Response: Simple summary was simplified to prevent overlapping with the introduction.

  1. In the part of materials and methods, 608 1-7 adult birds were selected as experimental materials. Whether it has some reference value for other commercial breeds or models aiming at obtaining the maximum economic value in a short time.

Response: We corrected it.

  1. The clarity of figure 1, 3, and 4 is low, please change them.

Response: Figure quality was improved and they were changed in the body text.

  1. Some of the references are formatted incorrectly. Please review them and modify them according to the format of Animals.

Response: We revised the bibliography as suggested by the reviewer.

Reviewer 2 Report

This is very interesting manuscript, with proper methodology and interpretation of the results.

Just some minor issues which should be clarified:

L21 As You known, hen generally indicate adult female - you have both males and females in your study.

L22 473 hens and 135 roosters

L30 Name these breeds (Sureńa and Utrerana) and indicate their origin (Mediterranean trunk of Spanish autochthonous hens)

L32 Explain VIF abbreviation (variance inflation factor)

Table 1 In general al physical quantities should be defined with their corresponding units. Also consider changing “How to measure it” to “"measuring procedure" or "how it was measured"

Figure 2 Description of SL, SW, is missing; WW is marked in figure as LL.

L145 Any scale for measurements of color ?

L150 Which normality test was used ?

L185 It is required for journal, to give the location of manufacturers and providers of all software used in the analyses (SPSS, IBM, Armonk, NY, USA; XLSTAT, Addinsoft, Paris, France).

L212 It was tested using chi-squared test, right?

L217 Trace Criterion – lowercases

L221 What software was used for PCA?

L233-234 superscript, subscript where appropriate

L255 What software was used to perform CHAID?

L259 Chi-squared

Table 3 While abbreviations presented in Table are generally well known, they should be explained in table footnote (see Table 6)

L301-302 Supplementary Table S1 and S2 show only variables with VIF <5, i.e. included in the analysis. Please correct the description in the text.

L311 90.71% or 90.37% (see table 5) ?

L314 You have only three function, this the short information that all selected functions explained over 95% of the total variance for both sexes could be given.

Figure 3. Change commas to dots in y-axes. (I know what this graph shows and how to interpret presented data, but consider improving its readability).

L378 “…were equal to or below 0.986 (Sup. Figure S3).”

L383 “… were equal to or below 1.015 (Sup. Figure S4).”

L438 meat is not “skinned”

Thank for the opportunity of reviewing this interesting work.

Author Response

REVIEWER 2

This is very interesting manuscript, with proper methodology and interpretation of the results.

Response: We thank the reviewer for his/her kind comments.

Just some minor issues which should be clarified:

L21 As You known, hen generally indicate adult female - you have both males and females in your study.

Response: We agree with the reviewer and changed hen to breed.

L22 473 hens and 135 roosters

Response: Added.

L30 Name these breeds (Sureńa and Utrerana) and indicate their origin (Mediterranean trunk of Spanish autochthonous hens)

Response: Added.

L32 Explain VIF abbreviation (variance inflation factor).

Response: Explained.

Table 1 In general al physical quantities should be defined with their corresponding units. Also consider changing “How to measure it” to “"measuring procedure" or "how it was measured"

Response: Added and changed.

Figure 2 Description of SL, SW, is missing; WW is marked in figure as LL.

Response: Added and modified.

L145 Any scale for measurements of color?

Response: No, we did not as colours were those reported in the standard of the breed.

L150 Which normality test was used?

The Shapiro–Francia W’ test (for 50 < n < 2500 samples) was used to discard gross violations of normality assumption. The Shapiro–Francia W’ test was performed using the Shapiro–Francia normality routine of the test and distribution graphics package of the Stata Version 15.0 software.

L185 It is required for journal, to give the location of manufacturers and providers of all software used in the analyses (SPSS, IBM, Armonk, NY, USA; XLSTAT, Addinsoft, Paris, France).

Response: Added.

L212 It was tested using chi-squared test, right?

Response: We clarified in the body text.

L217 Trace Criterion – lowercases

Response: Changed.

L221 What software was used for PCA?

Response: Changed.

L233-234 superscript, subscript where appropriate

Response: Changed.

L255 What software was used to perform CHAID?

Response: Tree routine of the Classify package of SPSS v26.0 software (IBM, Armonk, NY, USA).

L259 Chi-squared

Response: Added.

Table 3 While abbreviations presented in Table are generally well known, they should be explained in table footnote (see Table 6)

Response: Added.

L301-302 Supplementary Table S1 and S2 show only variables with VIF <5, i.e. included in the analysis. Please correct the description in the text.

Response: Corrected.

L311 90.71% or 90.37% (see table 5) ?

Response: Added.

L314 You have only three function, this the short information that all selected functions explained over 95% of the total variance for both sexes could be given.

Response: The rest of functions were not included given they resulted to be statistically non-significant.

Figure 3. Change commas to dots in y-axes. (I know what this graph shows and how to interpret presented data, but consider improving its readability).

Response: Changed. We clarified what the graph represents in Figure caption to improve its understandability.

L378 “…were equal to or below 0.986 (Sup. Figure S3).”

Response: Added.

L383 “… were equal to or below 1.015 (Sup. Figure S4).”

Response: Added.

L438 meat is not “skinned”

Response: We corrected it.

Thank for the opportunity of reviewing this interesting work.

Response: Thank you for your attention to our manuscript.

Reviewer 3 Report

The paper is interesting for the originality and the importance of biodiversity in poultry production.

My suggestions are:

row 22: 135 males?

row 41:, respectively?

row 114: from 1 to 7 years old: equally distributed among sex, varieties, ...?; you should add details.

row 129: you should add the date and n. of the response of Ethics Committee.

row 156-164: some traits are cited as index (check table 1 and table 2).

row 311: 90.71%? anyway, I suggest 1 decimal.

row 362: Press' Q (check at row 247-248)

row 438: from dark skin?

row 459: eliminate in which

row 474-478: this sentence is too much long.

When you consider the antipredator  behaviour, could the plumage colour  be considered also, in your opinion? 

Some typing mistakes in the text, check it.

Author Response

REVIEWER 3

The paper is interesting for the originality and the importance of biodiversity in poultry production.

Response: We thank the reviewer for his/her kind comments.

My suggestions are:

row 22: 135 males?

Response: Changed.

row 41:, respectively?

Response: We modified it.

row 114: from 1 to 7 years old: equally distributed among sex, varieties, ...?; you should add details.

Response: Details were added.

row 129: you should add the date and n. of the response of Ethics Committee.

Response: As it was added to the exemption letter forwarded to the editorial office according to the premises described in the below mentioned rules and regulations, the present study is out of the scope of evaluation of the below mentioned committee, hence no consultation was needed.

row 156-164: some traits are cited as index (check table 1 and table 2).

Response: The word index was changed to the word ratio.

row 311: 90.71%? anyway, I suggest 1 decimal.

Response: Corrected. We decided to leave it with two decimals to make it agree with the rest of numbers provided in the manuscript.

row 362: Press' Q (check at row 247-248)

Response: Corrected.

row 438: from dark skin?

Response: Corrected.

row 459: eliminate in which

Response: We eliminated it.

row 474-478: this sentence is too much long.

Response: We agree with the reviewer. Sentence was divided into three smaller sentences.

When you consider the antipredator behaviour, could the plumage colour  be considered also, in your opinion? 

Response: Feather coloration strongly conditions the camouflage abilities of birds. In these regards, Dohner [1] suggested the less aggressive strains developed for confinement may be less self-sufficient and may not be as alert to predators. In hens, this has been ascribed to the association of specific quantitative trait locus with behavioral traits [2]. As an example, to depict this, birds carrying the ancestral junglefowl allele (i) of the PMEL17 locus are black, while, White Leghorn (I) are white (with heterozygotes frequently being less pigmented). 

Contextually, i/i alleles carriers have been reported to be more vocal, less prone to develop fearful attitudes towards humans, and more aggressive, social and explorative (enhanced foraging behavior) [2]. These enhanced behavioral features may make these dark-colored breeds less susceptible to predation by hawks [1]. PMEL17 locus has simultaneously been associated with feather-pecking and bullying behavior towards counterparts [3], with darker birds tending to be rather affected by feather-pecking than their white counterparts [4]. It still unknown whether, feather-pecking may exclusively be attributed to plumage color or to the behavior of i/i carrier individuals to become targets of pecking attacks. 

Alternatively, Tickell [5] stated that coloration related costs in higher rates of bird predation may also translate into the enhancement of other tactics for evading capture [6]. This was reflected in our study (Figures 5 and 6) with Sureña presenting smaller ocular indexes in comparison to Utrerana hens, with however, darker Sureña individuals being closer to white Utrerana animals, and white feathered Sureña locating further away when morphological traits are considered.

  1. Dohner, J.V. The encyclopedia of historic and endangered livestock and poultry breeds; Yale University Press: New Haven, CT, USA, 2008.
  2. Wiener, P.; Wilkinson, S. Deciphering the genetic basis of animal domestication. Proceedings of the Royal Society B: Biological Sciences 2011, 278, 3161-3170.
  3. Keeling, L.; Andersson, L.; Schütz, K.; Kerje, S.; Fredriksson, R.; Carlborg, Ö.; Cornwallis, C.; Pizzari, T.; Jensen, P. Chicken genomics: Feather-pecking and victim pigmentation. Nature 2004, 431, 645-646.
  4. Nätt, D.; Kerje, S.; Andersson, L.; Jensen, P. Plumage Color and Feather Pecking—Behavioral Differences Associated with PMEL17 Genotypes in Chicken (Gallus gallus). Behavior Genetics 2007, 37, 399-407, doi:10.1007/s10519-006-9125-0.
  5. Tickell, W.N. White plumage. Waterbirds 2003, 26, 1-12.
  6. White, C.M.; Weeden, R.B. Hunting Methods of Gyrfalcons and Behavior of Their Prey (Ptarmigan). The Condor 1966, 68, 517-519, doi:10.2307/1365332.

Some typing mistakes in the text, check it.

Response: The manuscript was revised and checked for typos and grammar inconsistencies by a Cambridge ESOL examination instructor.